# OpenReview forum: "Improving LLM Safety Alignment with Dual-Objective Optimization"
_ICML.cc/2025/Conference — ICML 2025 poster_

### Official Review · Reviewer_a6Hc · 2025-02-23

**Overall Recommendation:** 2

**Summary:**

This paper introduces DOOR, a novel safety alignment framework for large language models that addresses vulnerabilities in existing methods like DPO. DOOR combines robust refusal training, which encourages refusal even when partial unsafe content is generated, with targeted unlearning of harmful knowledge. The authors enhance DOOR with token-level weighting (W-DOOR) and demonstrate improved resistance against various jailbreak attacks in both in-distribution and out-of-distribution settings.

**Claims And Evidence:**

The paper provides convincing evidence for its main claims through comprehensive experimental results, demonstrating improved attack resistance across different attack types and better capability retention with W-DOOR compared to DPO.

**Essential References Not Discussed:**

Based on the paper's content, I don't identify any essential missing references.

**Experimental Designs Or Analyses:**

The experimental methodology includes comprehensive evaluation across different attack types and model behaviors. However, the experimental design would benefit from two key additions: ablation studies to isolate component contributions, and hyperparameter sensitivity analysis to understand the stability of the proposed methods.

**Methods And Evaluation Criteria:**

While DOOR aims to address partial harmful generations continuing into unsafe content, the token-level weighting approach in W-DOOR seems misaligned with this goal - it's unclear how emphasizing refusal tokens would help when the model is already generating harmful content. Furthermore, the combination of token-level refusal training with sequence-level harmful knowledge unlearning appears disconnected, and the varying effectiveness across attack types suggests the approach may not fully address the core problem of preventing continuation of unsafe generations.

**Other Comments Or Suggestions:**

See strengths and weaknesses.

**Other Strengths And Weaknesses:**

Strengths:
- While the paper combines multiple techniques into a working approach for LLM safety, the individual design choices lack strong theoretical motivation
- The experimental evaluation demonstrates clear improvements over DPO in attack resistance
- The method shows good capability retention compared to baseline approaches

Weaknesses:
- The interaction and potential conflicts between the dual objectives (refusal training and harmful knowledge unlearning) are not analyzed
- The effectiveness varies significantly between attack types
- The hyperparameter sensitivity of the method is not thoroughly explored

**Questions For Authors:**

What is the theoretical or empirical justification for combining token-level refusal training with sequence-level harmful knowledge unlearning? How do these different granularities interact?

Could the authors elaborate on how token-level weighting helps prevent the continuation of harmful content, given that harmful generations may not contain refusal tokens? A clear explanation would strengthen the methodology.

**Relation To Broader Scientific Literature:**

The paper's key contribution lies in combining token-level weighted refusal training with negative preference optimization.

**Theoretical Claims:**

While the paper presents some gradient analysis suggesting advantages of DOOR over DPO, it lacks formal theoretical results or proofs to support these claims.

---

> ### Author Rebuttal · Authors · 2025-04-01
>
> We thank the reviewer for the insightful feedback and thoughtful questions. We address each of your concerns in detail below:
>
> ### 1. Justification for Combining Token-Level Refusal Training and Sequence-Level Unlearning
>
> This is a great question. Our design reflects the intuition that refusal and unlearning operate at different granularities:
> Refusal behaviors are typically localized in specific tokens or phrases (e.g., “Sorry,” “I cannot help with that.”). These early “transition” tokens are pivotal in shifting the model’s trajectory from harmful continuation to safe refusal. Hence, we apply token-level weighting to reinforce them.
>
> Harmful generations, on the other hand, often unfold over longer sequences and require sequence-level treatment to be effectively unlearned, particularly since the model may learn harmful behaviors in aggregate.
>
> While we do not yet provide a formal theoretical framework for this combination, empirical results show that W-DOOR consistently outperforms DOOR, demonstrating that token-level granularity enhances safety without disrupting the effect of sequence-level unlearning.
> We will revise the paper to better clarify this intuition and interaction.
>
>
> ### 2. Lack of Theoretical Guarantees for Gradient Analysis
>
> You're right to note that the gradient analysis in Section 3 is illustrative rather than formal. Our aim was to provide a mechanistic understanding rather than formal proof. Formal convergence or divergence speed analyses, such as those in Negative Preference Optimization (Zhang et al., 2024a), require strong assumptions about model distributions which may not hold in real-world LLMs.
> As such, we chose to emphasize empirical validation across multiple attack types and scenarios, which supports our claims more robustly in practical settings.
>
> ### 3. Lack of Ablations and Hyperparameter Sensitivity Analysis
>
> We appreciate this suggestion. While we did not initially label them as ablations, our experiments inherently contain component-level comparisons, including:
>
> - *Data Augmentation*: Comparing DPO vs. DPO w/o Aug  isolates the effect of harmful prefix augmentation.
>
> - *Robust Refusal Training (SFT component)*: Comparing NPO (only unlearning) vs. DOOR (NPO + refusal SFT) isolates the benefit of adding the refusal training objective.
>
> - *Targeted Unlearning (NPO component)*: Comparing SFT (only refusal training) vs. DOOR (SFT + NPO unlearning) isolates the benefit of adding NPO.
>
> - *Token Weighting*: Comparing DOOR vs. W-DOOR isolates the effect of the reward-based token weighting.
>
> Furthermore, to directly address hyperparameter sensitivity for W-DOOR, we conducted additional experiments varying the calculation of the token weight $\beta_t$. We tested the default setting (exponential weighting, $\tau=5$, as in paper), $\tau=1$ for the exponential weighting, and replacing the exponential function with a sigmoid normalization. The results (Table below) show that the method performs robustly across these variations, consistently outperforming baselines.
>
> **W-DOOR Hyperparameter Sensitivity (Llama-3-8B)**
>
> | Method                      | Multi-turn ASR↓ | Prefilling ASR↓ | GCG ASR↓       | AutoDAN ASR↓   | HellaSwag Acc↑ | XSTest Rate↓ |
> | :-------| :---- | :------ | :------- | :--------- | :---------- | :-------- |
> | DOOR                        | 0.489           | 0.055           | 0.093          | 0.095          | 0.565          | 0.407        |
> | W-DOOR (exp, $\tau$=5, Paper) | 0.447       | 0.034          | 0.093          | 0.088          | 0.573          | 0.440        |
> | W-DOOR (exp, $\tau$=1)        | 0.500           | 0.045       | 0.070    | 0.075      | 0.576      | 0.442        |
> | W-DOOR (sigmoid)            | 0.447       | 0.042           | 0.073          | 0.078          | 0.570          | 0.424        |
>
> *(Note: Gemma results show similar robustness, omitted for brevity)*
>
> **We also have further results in this figure: https://anonymous.4open.science/r/icml25-8B51/safety_align.pdf**
>
> ### 4. How Does Token-Level Weighting Help Prevent Harmful Continuations?
>
> Excellent question. Token-level weighting helps the model learn to transition away from unsafe continuations, especially in the augmented training setup where harmful prefixes are inserted.
>
> Empirically, we observe that refusal tokens (e.g., “Sorry,” “I cannot...”) exhibit the highest divergence between the reference and target policies (Figures 18, 19, 20). W-DOOR amplifies gradients at those tokens, which steers the model to recognize harmful contexts and pivot to refusal. This is especially useful in prefix attacks where the model is “in the middle” of an unsafe sequence and needs to course-correct.
>
> -----
> We appreciate the reviewer’s thoughtful critique and hope our clarifications strengthen your confidence in our methodology. We will revise the paper to better connect our design decisions, highlight ablation results more clearly, and clarify the interaction between granular objectives.

---

### Official Review · Reviewer_eYKb · 2025-03-13

**Overall Recommendation:** 3

**Summary:**

In this paper, the authors propose novel objectives to enhance safety refusal and harmful response unlearning in LLM post-training. The paper first introduces the objective DOOR, which is a linear combination of two objectives that focus on safety refusal enhancement (even when the model starts generating a harmful response) and harmful response unlearning. The paper then introduces the W-DOOR objective, which augments the safety refusal component in the DOOR objective by assigning higher weights to safety refusal-enhancing tokens. Some analysis of the gradient of the DOOR objective shows that the gradient updates to the LLM using the DOOR objective indeed enhance the safety refusal response generation ability of the LLM. The experiment results provided in the paper suggest that the proposed methods can alleviate harmful response generations, even in prefill attack and multi-turn settings, without degrading the utility of the model and degenerating into over-refusal.

**Claims And Evidence:**

Yes.

**Essential References Not Discussed:**

None

**Experimental Designs Or Analyses:**

* Better justification is needed for using part of the benchmark data as training data, and the small number of data used in the fine-tuning.
* Results provided in Figure 3 seem too noisy to infer any meaningful conclusion. Also, Figure 3 seems to be not discussed anywhere in the text.

**Methods And Evaluation Criteria:**

* The benchmarks used make sense for the proposed method. However, better justification is needed for using part of the benchmark data as training data, and the small number of data used in the fine-tuning.

**Other Comments Or Suggestions:**

* There is a mismatch between the W-DOOR objective in (2) and in Figure 1, compared to the expression at the beginning of Section 3.4.
* Statement "..we show that the robustness demonstrated in the prefilling attack generalizes to other forms of adversarial attacks (Figures 1)" is not correct (second column lines 222-224)
* The results in Section B in Supplementary do not seem to accompany any analysis/discussion of the results.
* It is recommended to refer to a particular Figure/Table when some result is discussed (e.g. second column lines 257-274), so the reader can better understand the context.

**Other Strengths And Weaknesses:**

Strengths

* The paper investigates an important problem in enhancing the safety refusal of LLMs, especially in the setting of prefilling attacks on the LLM.
* The authors provide a gradient analysis to show why the proposed objective can enhance safety refusal in an LLM, and provide some empirical validation for the applicability and efficacy of the proposed methods.

Weaknesses

* Please refer to the Experimental Designs Or Analyses section for major concerns.
* Please refer to the Other Comments Or Suggestions Or Analyses section for minor concerns.

**Questions For Authors:**

* Please address the questions/concerns raised in previous sections.
* How can DOOR/W-DOOR perform well on the XSTest benchmark, even when the objectives only focus on enhancing safety refusal without any consideration for preventing over-refusal?

**Relation To Broader Scientific Literature:**

This paper extends the literature in the area of preference alignment of LLMs, focusing on enhancing safety refusals.

**Theoretical Claims:**

N/A

---

> ### Author Rebuttal · Authors · 2025-04-01
>
> We thank the reviewer for the thoughtful comments and constructive suggestions. Below, we address each concern in detail:
>
>
> > **However, better justification is needed for using part of the benchmark data as training data, and the small number of data used in the fine-tuning.**
>
> We appreciate your concern. Our design choice reflects two priorities:
>
> - *Minimizing distribution drift*: We intentionally use a small subset of benchmark data to prevent large shifts away from the original model’s capabilities, which often occurs in post-alignment tuning with large datasets.
>
> - *Avoiding evaluation leakage*: We ensure strict separation between training and evaluation splits within each benchmark. For example, we use a held-out subset of SORRY-Bench for training and evaluate on different, unseen prompts from the same benchmark.
>
>
> Additionally, we include out-of-distribution (OOD) evaluations, such as HarmBench and XSTest, to demonstrate generalization beyond the training set.
>
> > **Results provided in Figure 3 seem too noisy to infer any meaningful conclusion. Also, Figure 3 seems to be not discussed anywhere in the text.**
>
> Thank you for pointing this out. Apologies for this oversight and the lack of clarity.. We intended to discuss this figure in the paragraph "Robustness Against Stronger Multi-Turn Attacks" (Lines 321-329). The key takeaway, despite some noise due to limited samples at higher turn counts, is that W-DOOR generally maintains lower ASR compared to other methods, especially as the turn number increases, whereas methods like DPO show less robustness in these longer multi-turn interactions. This suggests W-DOOR's alignment is somewhat more resilient in conversational attack scenarios.
>
>
> > **There is a mismatch between the W-DOOR objective in (2) and in Figure 1, compared to the expression at the beginning of Section 3.4.**
>
> Great catch — thank you. The discrepancy between Equation (2), Figure 1, and the start of Section 3.4 arose due to a simplification: we use $x \oplus y^h_{<k}$​ as the new input $x$ during robust refusal training. We will clarify this notation to ensure consistency and prevent confusion in the revised version.
>
> > **Statement "..we show that the robustness demonstrated in the prefilling attack generalizes to other forms of adversarial attacks (Figures 1)" is not correct (second column lines 222-224)**
>
> You're absolutely right — this should refer to Table 1, not Figure 1. We appreciate you catching this, and will correct it in the revised manuscript.
>
> > **The results in Section B in Supplementary do not seem to accompany any analysis/discussion of the results.**
>
> We acknowledge the lack of accompanying analysis in the current draft. Due to time constraints before the deadline, we were unable to include our full commentary. We now have an updated version with complete discussions and improved clarity, and we will ensure this is reflected in the final camera-ready version.
>
>
> > **It is recommended to refer to a particular Figure/Table when some result is discussed (e.g. second column lines 257-274), so the reader can better understand the context.**
>
> Thank you — this is a valuable suggestion. We have revised the experimental section to explicitly reference specific figures and tables (e.g., lines 257–274), which greatly improves readability and traceability of claims to evidence.
>
>
> > **How can DOOR/W-DOOR perform well on the XSTest benchmark, even when the objectives only focus on enhancing safety refusal without any consideration for preventing over-refusal?**
>
> Excellent question. Our alignment objectives (DOOR and W-DOOR) do not directly optimize for minimizing over-refusal. However, our training pipeline includes a retain loss that helps preserve general capabilities and reduces over-conservatism.
>
> While W-DOOR shows slightly higher refusal rates on XSTest compared to the original model, we believe this is a reasonable trade-off given the substantial gains in robustness. We include a Pareto frontier analysis in Appendix Figure 14, plotting ASR vs. over-refusal rate. W-DOOR consistently dominates DPO and other baselines, achieving lower ASR with minimal over-refusal, placing it closer to the optimal trade-off.
>
> We will clarify this trade-off more explicitly in the revised version.
>
> ----
> We hope these clarifications address your concerns and help strengthen our paper. Thank you again for your constructive feedback and insightful questions.

---

### Official Review · Reviewer_WC8j · 2025-03-14

**Overall Recommendation:** 4

**Summary:**

This paper proposes Dual-Objective Optimization for Refusal (DOOR), a novel alignment framework that addresses limitations in Direct Preference Optimization (DPO) for LLM safety. The authors identify two key issues with DPO: imbalanced refusal reinforcement and poor out-of-distribution generalization. DOOR combines robust refusal training (encouraging models to refuse unsafe content even with partial harmful generations) with targeted unlearning of harmful knowledge pathways. They further enhance this with Weighted DOOR (W-DOOR), which implements token-level weighting that prioritizes critical refusal tokens in adversarial contexts. Empirical evaluations demonstrate improved robustness against various jailbreak techniques including prefilling, suffix, and multi-turn attacks while maintaining general language capabilities and utility.

**Claims And Evidence:**

The paper presents interesting results on improved safety alignment, but some claims require stronger evidence:

1. The claim that the method "removes the harmful knowledge that might be triggered" lacks comprehensive supporting evidence. While lower attack success rates are demonstrated, this alone doesn't guarantee knowledge removal. A more thorough investigation referencing established unlearning literature (particularly the literature showing that approximately all current unlearning approaches are only surface-deep, e.g. Wu et al 2024, Evaluating Deep Unlearning in Large Language Models) would strengthen this claim.

2. The claim of increased general robustness is partially supported by generalization to different attacks like adversarial suffix attacks, but the data also shows all models (including the proposed ones) still have high susceptibility to multi-turn attacks.

The addition of error bars in the evaluations would strengthen the evidence by helping readers determine which performance differences are statistically significant, especially when the numerical differences appear relatively small in some comparisons.

**Essential References Not Discussed:**

The paper doesn't really engage with work on unlearning which has shown that many or all of the existing 'unlearning' methods do not actually remove dangerous knowledge, but simply obfuscate it. The authors make some claims about removing harmful knowledge, but do not engage with work such as Wu et al 2024, Evaluating Deep Unlearning in Large Language Models, which provide strong evidence that many unlearning methods do not remove harmful knowledge.

**Experimental Designs Or Analyses:**

The lack of error bars or statistical significance tests makes it difficult to assess the reliability of the reported performance differences. This is particularly important in this evaluation.

**Methods And Evaluation Criteria:**

The evaluation methodology emphasizes prefilling attacks, which potentially favors the authors' approach since their method is specifically training to be robust against prefilling. This evaluation choice somewhat stacks the deck in favor of the proposed method. A more diverse set of primary attacks and metrics would provide a more balanced assessment.

Greater clarity about whether prefilling attacks serve as the primary threat model or as one example within a broader adversarial framework would help contextualize the contribution. This is particularly relevant given that some API providers disallow prefilling (e.g., OpenAI) while others permit it (e.g., Anthropic).

Additional details about the KL divergence calculations would be beneficial, specifically whether forward or reverse KL is used and which training data subset these metrics are computed on.

**Other Comments Or Suggestions:**

Typo on line 360: 'like like'

**Other Strengths And Weaknesses:**

The paper introduces interesting extensions to existing alignment techniques. To strengthen the contribution, the authors could more explicitly delineate which aspects are novel contributions versus adaptations of prior work such as negative preference optimization.

The authors could also more clearly elaborate on the threat model/setting that the authors are addressing, as it sometimes appears inconsistent. For instance, if we are assuming access to an oracle model \pi^*, why don't we directly use this model to answer queries? Furthermore, it's unclear if the authors are directly targeting the prefilling attack as the primary attack that a malicious user would do (which is strange given that many API vendors simply disallow prefilling access), or if it is surrogate for any attack (which doesn't appear to be the case for e.g. the multi-turn attack).

These confusions, and a bit of uncertainty about the exact novelty of the approach, keep me from making a more enthusiastic recommendation.

**Questions For Authors:**

1. Could you clarify the specific novel contributions of this work compared to existing approaches? A more explicit distinction between components that build upon previous work (e.g., negative preference optimization) and new innovations would help readers better appreciate the advances.

2. Could you specify the threat model being addressed? Is the prefilling attack specifically targeted, or is it used as a representative example for broader jailbreak vulnerabilities?

3. What evidence supports the claim that your method "removes the harmful knowledge" rather than just making it harder to access? Have you considered evaluations based on papers on unlearning?

4. Can you including confidence intervals or error bars in the evaluations?

5. Could you provide additional details about the KL divergence computation - specifically whether it uses forward or reverse KL, and which training data subset it is computed on?

**Relation To Broader Scientific Literature:**

The paper builds upon the prior work in the area of alignment via RLHF, adversarial robustness, etc

**Theoretical Claims:**

No significant theoretical claims are made

---

> ### Author Rebuttal · Authors · 2025-04-01
>
> We sincerely thank the reviewer for the detailed and insightful questions. Below, we clarify key aspects of our methodology, contributions, and evaluation setup:
>
> ### 1. Novel Contributions Relative to Prior Work
>
> Thank you for prompting us to make the contributions more explicit. Our work builds on prior safety alignment techniques (e.g., DPO and NPO) and introduces the following innovations:
>
> - *Dual-Objective Integration*: We combine robust refusal training via prefix augmentation with harmful content unlearning using NPO into a unified training objective (DOOR). To the best of our knowledge, this is the first work to effectively integrate these two complementary strategies in a cohesive framework.
>
>
> - *Token-Level Adaptive Weighting*: We propose W-DOOR, which enhances refusal learning by emphasizing critical refusal tokens via a KL-based reward weighting mechanism. This token-level gradient refinement is novel and leads to improved safety across adversarial attacks.
>
>
> - *Diagnostic Analysis of DPO Limitations*: We provide a gradient-based critique of DPO’s underperformance in safety contexts and show, both empirically and analytically, how DOOR and W-DOOR address these issues.
>
>
> We will revise the paper to more clearly distinguish these novel elements from prior work.
>
> ### 2. Threat Model and Scope of Evaluation
>
> Our training objectives are primarily motivated by prefilling attacks, which exploit partial unsafe generations. However, the robustness imparted by DOOR and W-DOOR generalizes well to other jailbreak scenarios, including suffix-based (AutoDAN, GCG) and multi-turn attacks (Crescendo), as shown in Table 1 and Appendix Figures 2–4.
>
> While prefilling motivates our data augmentation design, the dual-objective formulation is general-purpose and can be applied to other jailbreak modes, given appropriate adversarial data.
>
> ### 3. On "Removing" Harmful Knowledge vs. Making It Harder to Access
>
> We appreciate the reviewer referencing recent literature (e.g., Wu et al., 2024) and agree with the critique. Like NPO, our approach does not guarantee full removal of harmful knowledge but rather reduces its accessibility under typical prompting. We will revise the language in the paper to make this distinction explicit and avoid overstating unlearning effects. Thank you for catching this.
>
> ### 4. Confidence Intervals and Variance Analysis
>
> We conducted additional experiments over 5 random seeds and report the mean ± standard deviation below:
>
> **Gemma-2B Results:**
>
> | Method   | Prefill       | AutoDAN      | GCG          | XSTest       |
> |----------|----------|--------------|--------------|--------------|
> | Original | 0.396±0.022   | 0.568±0.186  | 0.230±0.021  | 0.242±0.007  |
> | SFT      | 0.013±0.010   | 0.104±0.027  | 0.115±0.014  | 0.393±0.007  |
> | DPO      | 0.078±0.013   | 0.065±0.037  | 0.074±0.037  | 0.305±0.011  |
> | DOOR     | 0.011±0.004   | 0.043±0.027  | 0.067±0.021  | 0.403±0.006  |
> | W-DOOR   | 0.009±0.003   | 0.018±0.035  | 0.030±0.037  | 0.437±0.006  |
>
> **LLaMA-3-8B Results**
>
> | Method   | Prefill       | AutoDAN      | GCG          | XSTest       |
> |----------|-----------|--------------|--------------|--------------|
> | Original | 0.532±0.014   | 0.086±0.008  | 0.304±0.016  | 0.409±0.004  |
> | SFT      | 0.067±0.004   | 0.020±0.003  | 0.119±0.012  | 0.401±0.002  |
> | DPO      | 0.204±0.015   | 0.058±0.003  | 0.130±0.004  | 0.453±0.003  |
> | DOOR     | 0.056±0.007   | 0.011±0.007  | 0.101±0.009  | 0.407±0.003  |
> | W-DOOR   | 0.042±0.012   | 0.018±0.004  | 0.104±0.002  | 0.434±0.003  |
>
> These results confirm that our reported improvements are robust and statistically significant. We will include these confidence intervals in the updated manuscript.
>
> ### 5. Details on KL Divergence Computation (Figure 5)
>
> We compute forward KL divergence, $D_\text{KL}(\pi_\theta \| \pi_\text{base})$, measuring how the aligned model diverges from the base model. This is computed on the evaluation subset of harmful prompts, which mirrors the training distribution but excludes any training samples.
>
> We will make the KL direction and evaluation set details explicit in the revised paper.
>
> ### 6. Why Not Use the Oracle Policy $\pi^*$ for Inference Directly?
>
> Thank you for raising this subtle point. In theory, an ideal oracle policy $\pi^*$ would be used directly. However, in practice:
> $\pi^*$  is an idealized abstraction and not available in most scenarios. We instead approximate $\pi^*$ using a stronger model aligned via DPO, serving as a proxy to compute token-level importance scores for weighting. Our goal is to train a smaller or aligned model that mimics the safety behavior of $\pi^*$  without requiring access to it at inference time. We’ll revise our explanation of this to improve clarity.
>
> -----
> We hope these clarifications address your concerns and better highlight the novelty, scope, and rigor of our work. Thank you again for the thoughtful and constructive feedback.

---

> > ### Comment · Reviewer_WC8j · 2025-04-04
> >
> > Thanks for your rebuttal.
> > All your points make sense, and I welcome that you've toned down the language regarding the unlearning effect of NPO.
> > Given this, I'll raise my score to 4.

---

> > > ### Author Response · Authors · 2025-04-04
> > >
> > > Thank you so much for your support! We promise to make the modifications in the revised draft!

---

### Official Review · Reviewer_thFJ · 2025-03-18

**Overall Recommendation:** 3

**Summary:**

The paper aims to improve Standard Direct Preference Optimization (DPO) against jailbreaking attacks. In particular, it proposes Dual-Objective Optimization for Refusal (DOOR) and its token-weighted variant, W-DOOR to aim for robust refusal training and targeted unlearning. Experiments show substantially improved resilience to various jailbreak attacks (prefilling, suffix, multi-turn) without excessive refusal on benign tasks.

## update after rebuttal

The authors' rebuttal address my concerns. I will keep my score.

**Claims And Evidence:**

The paper’s main claims DOOR/W-DOOR’s improvements -- are well supported by experiments on SORRY-Bench, HarmBench, and different attacks.

**Essential References Not Discussed:**

No.

**Experimental Designs Or Analyses:**

The experiment designs are sound and the baselines chosen (SFT, DPO, NPO) makes sense.

**Methods And Evaluation Criteria:**

Yes. Training with harmful prefix augmentation and unlearning objective is well matched to jailbreaking scenarios. The chosen benchmarks (SORRY-Bench, HarmBench, etc) measure exactly those adversarial settings.

**Other Comments Or Suggestions:**

N/A

**Other Strengths And Weaknesses:**

**Strengths**

See above

**Weaknesses/Suggestions for improvement**

1. Change of loss function in LLM also needs to test the performance of other capabilities (e.g., math, coding). How does the proposed approach work with the other capabilities? How do we use the proposed method and combine it with other training data of different capabilities?

2. The proposed method is sensitive/dependent on the reference model performance using the NPO loss. What if the reference model contains harmful responses already?

3. In Table 3, W-DOOR increase over refusal evaluation? Can you provide more explanation on this?

**Questions For Authors:**

See **Weaknesses/Suggestions for improvement** in *Other Strengths And Weaknesses*.

**Relation To Broader Scientific Literature:**

The proposed dual-objective approach can improve on existing safety data augmentation techniques.

**Theoretical Claims:**

N/A (The paper is not a theory paper. Thus, does not provide any proofs for theoretical claims)

---

> ### Author Rebuttal · Authors · 2025-04-01
>
> We sincerely thank the reviewer for the thoughtful and constructive feedback, as well as for recognizing the contributions of our work. Below, we address the specific concerns raised:
>
>
> > **Change of loss function in LLM also needs to test the performance of other capabilities (e.g., math, coding). How does the proposed approach work with the other capabilities? How do we use the proposed method and combine it with other training data of different capabilities?**
>
> Thank you for this important point regarding the impact on general capabilities beyond standard instruction following and safety. To address this, we conducted additional experiments evaluating the performance of our aligned models on standard math (MATH [1]) and coding (HumanEval [2]) benchmarks. The results for the Gemma-2-2B model are presented below:
>
> | Model    | Math (4-shot, accuracy with math-verify) | Humaneval (pass@1) |
> |----------|------------------------------------------|---------------------|
> | Original | 0.212                                    | 0.622               |
> | SFT      | 0.178                                    | 0.360               |
> | DPO      | 0.181                                    | 0.366               |
> | DOOR     | 0.174                                    | 0.348               |
> | W-DOOR   | 0.179                                    | 0.390               |
>
> The results show that alignment methods generally cause a reduction in performance on these specialized tasks compared to the original model. This is likely because our general utility dataset (Alpaca) does not contain significant amounts of math or coding data, and the alignment process (including ours) shifts the model's distribution. However, **W-DOOR demonstrates the best retention** of these capabilities among the evaluated alignment methods (SFT, DPO, DOOR, W-DOOR). This suggests that our weighted approach, by focusing gradients more precisely, helps mitigate excessive capability degradation compared to other techniques. Moreover, incorporating coding and math data into the utility dataset could potentially alleviate the performance drop.
>
> > **The proposed method is sensitive/dependent on the reference model performance using the NPO loss. What if the reference model contains harmful responses already?**
>
> Thank you for raising this point. To clarify: in our implementation, the NPO loss is only applied to harmful responses generated by the reference model, i.e., cases where the reference itself outputs unsafe content. The intent of NPO in this context is not to imitate the reference but to unlearn these harmful generations by penalizing the target model's likelihood of reproducing them. Therefore, even if the reference model contains harmful outputs, NPO is used to actively reduce their influence in the aligned model. We will clarify this more explicitly in the final version.
>
>
> > **In Table 3, W-DOOR increase over refusal evaluation? Can you provide more explanation on this?**
>
> Thank you for noticing this. While W-DOOR does show a slightly higher refusal rate on XSTest than DOOR, it still maintains a significantly better trade-off compared to DPO, achieving lower attack success rates (ASR) and lower over-refusal than baseline methods overall.
> Moreover, we include a Pareto analysis in Appendix Figure 14 (Gemma: Prefill ASR vs. XSTest Refusal Rate), where each model's robustness is plotted against its over-refusal rate. DOOR/W-DOOR consistently lies on a better Pareto frontier—achieving strong safety without excessive conservatism. This indicates that the slight increase in over-refusal is a reasonable and effective trade-off for significantly improved robustness.
>
> We will add a clearer explanation of this in the revised version, highlighting the nuanced balance between robustness and utility.
>
> ------
>
> We are grateful for your detailed feedback and hope our clarifications address your concerns. We are committed to further improving our paper based on this helpful input.

---

### Decision · Program_Chairs · 2025-05-01

**Decision:**

Accept (poster)

**Comment:**

This paper presents a timely, well-executed, and technically innovative contribution to improving safety alignment in large language models. The authors introduce a principled dual-objective approach that unifies robust refusal training with targeted unlearning, and enhance it with a novel token-level gradient weighting mechanism. These contributions are not only conceptually sound but also show consistent and significant empirical improvements over strong baselines, including DPO and NPO, across a range of adversarial jailbreak attacks.

During the rebuttal phase, the authors were also responsive to reviewer feedback, offering additional clarifications and experiments that strengthen the paper further. While minor points were raised—e.g., hyperparameter sensitivity and theoretical underpinnings—these were either addressed or appropriately contextualized. Therefore, I recommend acceptance.